# A Narrative Review of ^99m^Tc-Aprotinin in the Diagnosis of Cardiac Amyloidosis and a New Life for an Unfairly Abandoned Drug

**DOI:** 10.3390/biomedicines10061377

**Published:** 2022-06-10

**Authors:** Carlo Aprile, Lorenzo Lodola

**Affiliations:** IRCCS Fondazione Policlinico San Matteo, Nuclear Medicine Unit, I-27100 Pavia, Italy

**Keywords:** cardiac amyloidosis, technetium aprotinin, recombinant aprotinin, 30-51 SS cyclic peptide: synthetic insulin fibrils, radioactive Gallium, PET

## Abstract

Several studies investigated the use of ^99m^Tc-labelled Aprotinin as an amyloid seeker some years ago. In vitro tests showed high binding affinity for several types of amyloid fibrils accompanied by an excellent specificity. Initial human studies demonstrated good accuracy in detecting cardiac involvement. Scintigraphy results were confirmed in a group of 28 endomyocardial biopsies. Unfortunately, clinical studies were halted because of a temporary suspension of the vector protein (Trasylol) and public health concerns over prion contamination of the bovine origin compound. To obviate these limitations, efforts have been made to label a recombinant Aprotinin with 99mTc, which exhibits the same affinity for h-insulin fibrils. With the aim of developing a PET tracer, the same recombinant protein was labeled with Gallium. The introduction of a bifunctional chelator did not affect fibril affinity. Finally, a synthetic peptidic fragment, the cyclic 30-51 SS, was synthetized. After direct technetium labeling, an impressive increase in affinity was demonstrated. This peptide appears to be a potential candidate for Gallium labeling through a bifunctional chelator for PET imaging.

## 1. Introduction

The advent of effective therapies for the most common types of amyloid cardiomyopathy in recent years has boosted interest in these rare diseases, with the development of new imaging tools ranging from MRI through bone scintigraphy tracers, to PET imaging with radiotracers used for Alzheimer amyloidosis. Cardiac amyloidosis is often misdiagnosed, or its recognition is delayed as a result of both physician- and/or disease-related circumstances. Advances in multi-parametric cardiac imaging, including cardiac echography, have led to a deeper understanding of the disease process and to the tracking of treatment responses. However, no single diagnostic approach is so far able to perform a differential diagnosis, an early diagnosis and—in line with rapidly expanding treatment regimens—to monitor responses and assist with the adjustment of treatment strategies.

When we started our investigation on labeled aprotinin in 1995, the tracer available for systemic amyloidosis was ^123^I-SAP, which cannot, however, detect cardiac amyloidosis [1]. ^99m^Tc-labeled aprotinin (TcA) was introduced into the nuclear medicine armamentarium during the eighties due to its high renal tubule uptake reflecting kidney function [2]. Several years later, the same compound was studied for the imaging of cardiopulmonary amyloidosis with promising results [3], although this was scarcely available outside the UK [1] because cardiac involvement was poorly visualized with ^123^I-SAP.

Aprotinin, a serine protease inhibitor, was commonly employed (as Trasylol) in cardiac surgery to prevent blood loss until the Blood Conservation Using Antifibrinolytics in a Randomized Trial (BART) study initiated in response to concerns about increased mortality associated with this agent induced the EMA to suspend marketing authorization in 2008 [4]. However, the Agency’s Committee for Medicinal Products for Human Use (CHMP) found that there were a number of problems with the way the BART study was conducted, and the EMA recommended lifting the suspension of aprotinin in 2012 following the publication of the final BART study and other clinical studies showing that the benefits of aprotinin outweigh its risks in restricted indications [5,6].

Even if the marketing authorization suspension did not directly affect its use as a radionuclide vector, the lack of raw material meant that ongoing clinical studies in Europe and Australia were halted and were not resumed after the suspension was lifted. Only three further clinical papers on the topic are available, from Japan.

## 2. Binding Mechanism and Specificity

Proteolytic remodeling of the amyloid precursor involving serine-proteases is a critical step in the formation of AL and ATTR amyloid fibrils [7,8]. We hypothesized that a serine protease inhibitor such as aprotinin could enrich amyloid formation and deposition sites, allowing their imaging [3].

However, in an elegant in vitro study, Cardoso et al. [9] were able to demonstrate that: (1) there is specific binding of ^125^I-aprotinin to different types of fibril such as h-insulin (Ka 2.9 × 0.37 µM^−1^}, TTR V30M, λ-BJP and Aβ (1-42) without interaction with amorphous precipitates and/or soluble fibril precursors; (2) thioflavin and Congo Red do not compete for binding, indicating a different interaction; and (3) aprotinin binding has two major components, namely its interaction with the β-structure elements of both fibrils and ligands and an electrostatic effect.

There are other reports regarding the specificity of Tc-labeled aprotinin, the lack of significant binding to EGFr tumours in a mouse model, and the low non-specific uptake in an experimental rat model of sterile or Staphylococcus-induced inflammation [10,11].

## 3. Radiopharmacokinetics

Two papers have investigated the kinetics of the radiopharmaceutical after i.v. administration in humans [12,13].

Blood clearance is rapid with a biexponential function, with only 15% of the i.d. still circulating in the blood of a subject with normal kidney function 30 min p.i., and a significant correlation was found between this parameter and creatinine clearance. Cumulative urinary excretion amounted to 8% during the first 6 h in one report and to 3.5% at 4 h in the other.

Liver uptake was not negligible, being higher for instance in comparison to the commonly employed renal agent ^99m^Tc-DMSA. Maximum uptake was observed 90 min p.i., after which, no significant changes ascribable to catabolism were noted.

Sojan et al. determined 84.0 ± 0.8% of urinary radioactivity to be ^99m^Tc pertechnetate and 10.0 ± 7.0% to be unchanged ^99m^Tc-Aprotinin. However, some differences are remarkable between this study and the Bellitto et al. report [14] that found that, as with other low m.w. proteins, TcA is taken up by the tubular cells, metabolized, and in part excreted as labeled degradation products with a molecular weight of approximately 1 kDa. There is no immediate explanation for this discrepancy; even if the kits employed are slightly different (direct Sn Cl_2_ in one kit and stannous pyrophosphate in the other), the same amounts of aprotinin and reducing agent were employed [15].

## 4. Endomyocardial Biopsy

Myocardial biopsy represents the gold standard for validating the results of an imaging test—even if this involves the risk of sampling error—and a sufficient number of biopsies is therefore mandatory to reduce the risk of a false negative result; additionally the Congo Red staining is not without problems [16].

Validation of the TcA cardiac scan by myocardial biopsy has been reported by four groups working separately [3,17,18,19,20] and the results are summarized in Table 1. The data from the study by Awaya et al. [20] refers only to planar scans—not only to ensure uniformity with the other data, but also due to the discrepancies between the planar and SPECT-CT results that they reported. In fact, five planar scans produced true positives in the five subjects with positive biopsies, while SPECT-CT produced false positives in three out of five patients with negative biopsies [20].

The negative biopsy group in the table refers not only to amyloidosis patients without cardiac involvement but also to different cardiac diseases such as idiopathic cardiomyopathy (n.2) and infiltrative desmin cardiomyopathy (n.2) [3,18].

## 5. Diagnosis, Risk Stratification, and Long-Term Follow-Up

### 5.1. Referring Centers and Technical Approach

Clinical studies have been performed in five centers: in Pavia, Copenhagen, Glasgow, Sydney, and Tokyo, respectively. The Adelaide group results meanwhile are devoted to the study of kinetics in normal subjects [13]. Some papers take into consideration the detection of amyloid deposits in any region of the body, while others focus on cardiac amyloidosis (CA) only. The control group is heterogeneous, in some reports comprising subjects in whom amyloidosis was later ruled out, while other control groups included subjects with known cardiac diseases that potentially might be responsible for pathological uptake. In addition, the diagnostic work-up leading to a diagnosis of CA is frequently not well described and follow-up times are not specified.

Therefore, the limited number of patients with CA and control subjects, as well as variable diagnostic workups and follow-up times, impeded a systematic review of the available cases and so only a narrative review was possible.

Usually, imaging was obtained about 90–100 min after i.v. injection of 250 to 900 MBq of TcA. A SPECT or SPECT-CT of the cardiac area in addition to a planar scan were performed systematically in only some studies. Cardiac uptake was generally scored visually.

### 5.2. Clinical Cases Reports

The largest series [19] involved 103 studies performed on 78 patients (47 male and 31 female, median age 59 years, range 16–83). Seventy-three were affected by the classical AL form and two presented as MGUS, while in three of these, the amyloid was localized in the neck. On the basis of ECG and US findings, these patients had been previously classified into four categories according to pre-test probability of myocardial involvement: A—no suspicion, B—suspected, C—high probability or clear evidence without signs of congestive heart failure (CHF), and D—as in C but accompanied by CHF.

At the end of the observation period (1564 months), 40 patients were still alive (total follow-up time 996 months, median 31.7 months, range 4.2–60.3 months), 32 had died (median follow-up 12 months, range 7 days–56.8 months), and 9 were lost to follow-up or not fully evaluable; of these last nine patients, however, two underwent positive myocardial biopsy and one had a localized form of laryngeal biopsy (negative scan), and were included in the final report.

Taking into account the final clinical outcome, there were 2 false negative cases and 1 false positive, 35 true negative, and 36 true positive cases. One false negative case, a 60-year-old male patient with a normal ECG voltage but a cross-sectional area of 14 cm, developed congestive heart failure and died 30 months after the scan. The second case referred to a 43-year-old female with liver, kidney, and spleen involvement and with normal echographic findings at the time of the scan, who died 11 months after the scan from acute lung edema; although no definite relationship could be determined between the fatal disease and amyloid involvement, this case was prudentially considered as a false negative.

The false positive case was a 65-year-old hypertensive patient with a serum and urine IgA monoclonal component with a CSA of 13.8 and a voltage-to-mass ratio (V/M) of 0.48; he later received a stem cell transplant, and no clear signs of cardiac involvement were detectable 17 months after the scan (Table 2).

In this series, TcA scans were able to detect myocardial accumulation in two patients without any suspicion of cardiac involvement (group A): one of these further developed congestive heart failure from which he died one year later, while the second patient died suddenly 1 month later of fatal cardiac arrhythmia.

In groups B and C, TcA scans ruled out cardiac involvement in four and two patients, respectively, and confirmed involvement in 8 and 10 patients for each group. Myocardial involvement was confirmed in all patients of group D. Therefore, if the two lost patients of the group D+ with clinical/instrumental evidence of cardiac deposits are included, an overall accuracy of 96% can be observed.

Schaadt et al. [17] studied 22 fully evaluable patients with suspected or known amyloidosis including two cases of familial amyloidosis and three with the localized form. They reported positive cardiac scans in seven patients. Five of these were symptomatic while two were asymptomatic at the time of the scan, later developing heart failure.

In the Han et al. report [18], 99mTcA uptake of the heart was negative in all 30 subjects who did not receive a final diagnosis of cardiac amyloidosis (17 control subjects and 13 amyloid patients with no clinical or echocardiographic evidence of cardiac amyloid). Scans were positive in five patients without false positives in the amyloidosis or control group.

Four of these five patients died from cardiac complications within 2 years of diagnosis (mean f.u. 11 mo., range 1–14). A median cardiac-to-background uptake ratio of 1.1 (range 0.9–1.4) was found in controls, which was significantly lower than in the cardiac amyloid group, with a median value of 2.0 (range 1.6–2.4; *p* = 0.0004).

No false positive or false negative cardiac scans were reported in either of the above studies [17,18].

In an Australian pilot study (10 patients with biopsy-proven primary amyloidosis), cardiac uptake was observed in three symptomatic patients and a false negative in one patient with non-invasive evidence suggestive of CA [21].

The results of the cited reports are summarized in Table 3 below. Only one paper from the Tokyo group is reported because it was difficult to eliminate duplicated patients present in the papers quoted here.

### 5.3. Sensitivity and Specificity

As with sensitivity, specificity is also high in the European reports, while it is less satisfactory in the Tokyo group experience [20]. In fact, in 10 subjects with suspected amyloidosis, planar scans were positive in four of the five subjects with myocardial involvement and negative in the remainder. On the other hand, SPECT-CT scans were positive in all five patients with amyloidosis, whereas three out of five patients without amyloid deposits were false positive in SPECT/CT imaging, as demonstrated by endomyocardial biopsy. In a previous paper by the same group in Japan [23] on 25 patients with known (n. 19) or suspected amyloidosis and 11 positive cardiac scans, a weak uptake was falsely positive. Two asymptomatic patients with positive scans later developed new myocardial symptoms.

In other words, the number of false positive cases increased with the use of SPECT/CT in comparison to the planar method. This is quite surprising, because in the reports by Schaadt et al. [17] and Han et al. [18] SPECT, even if not performed systematically in all subjects, did not also add false positive results to the control group. Even if no immediate explanation is apparent, a technical reason may account for this discrepancy. Liver uptake is not negligible, and photons emitted in close proximity to the heart blur in the myocardium during data acquisition. In addition, SPECT images can suffer from artifacts caused by respiratory motion. Both of these factors decrease the diagnostic accuracy of SPECT imaging—especially in the inferior wall and septum [24,25,26].

As far as specificity is concerned, in another study relating to the use of TcA in dialysis-related amyloidosis (DRA, 25 patients), no uptake was detected in patients with longstanding hypertension and myocardial hypertrophy [27].

In addition, specificity is supported by a robust biochemical basis of aprotinin interaction with amyloid fibrils, confirming the binding specificity with fibrils from different sources and not with amorphous precipitates or soluble protein [9].

### 5.4. Unanswered Questions Arising from Clinical Studies

In all the analyzed reports, sensitivity is high, with only sporadic false negative results. This is obviously a favorable finding, but another question remains unanswered—namely, how precociously a positive scan in a patient without or with equivocal clinical and instrumental findings can precede overt evidence of cardiac involvement. The reported case studies are not sufficient to fully answer this question; however, there are some positive indications. In three reports [17,18,19], disagreement between scan positivity and evidence of involvement tended to confirm the disease up to several months later.

A second unanswered question regards the amount of uptake and its prognostic significance—in other words, whether this reflects the amyloid burden leading to organ impairment. There is insufficient data available to answer this, even if there may be a tendency to observe higher visual scores in more compromised hearts—at least in the AL type [22].

As mentioned above, anecdotal cases alone are not sufficient to answer the question of whether TcA scans can monitor responses to therapy.

### 5.5. Limits of TcA

To date, despite high sensitivity and specificity linked to robust biochemical background, TcA suffers from three major limitations.

First of all, there is the limited availability of the vector molecule as a clinically licensed drug and the associated safety concerns about its bovine origin, which caused the interruption in its radiopharmaceutical use.

Another limitation is related to the relatively moderate myocardial/background uptake ratio. These relatively moderate values do not appear to reduce sensitivity, but an improvement could allow the detection of subtle deposits, leading to a more precocious diagnosis.

A further potentially limiting factor is the physiological liver uptake. After the kidney, which is the main accumulation and excretion organ, the liver exhibits a non-negligible uptake. Even if per-unit volume is approximately 7 and 14% of the renal value in mice and rats, respectively [10,28], the entire organ may be an important source of radioactivity—causing a significant spillover to the inferior myocardial wall and septum, accentuated by respiratory movements. As mentioned above, this aspect may constitute a potential risk factor in SPECT, leading to false positive results.

## 6. Safety

No adverse events were reported in either patients or control subjects following administration of 99mTc-Aprotinin. Despite the allergenic potential of aprotinin, it is interesting to note that the amount usually employed is largely inferior to the 3500 KIU usually contained in a multidose vial, while the amount suggested for testing the risk to allergic/anaphylactic reactions is equivalent to 10,000 KIU [29].

## 7. Dosimetry

Following the i.v. administration of 500–700 MBq, the estimated dose equivalent in healthy subjects has been estimated to be <10 mSv [17], which is in the same range as reported by Sojan et al. [13] of 1.4–2.0 mSv for a dose of 250 MBq and by Han et al. [18] of 1.6 mSv for a dose of 200 MBq.

## 8. Future Directions

As previously depicted in Section 5.5, despite promising initial results—unfortunately unconfirmed in supplementary clinical trials—there are intrinsic limitations that have hampered further developments. In the following subsections, we attempt to outline possible approaches to overcome these limitations. Although these labeled compounds have not yet been tested in vivo, in vitro stability and affinity results suggest a potential role in CA detection.

### 8.1. Recombinant Aprotinin

Despite the lifting of the suspension on aprotinin as a therapeutic drug (Trasylol), its availability is still restricted in many countries. In addition, because of public health concerns over potential contamination with infectious agents such as BSE, some countries have taken precautionary steps regarding bovine-sourced materials in the manufacture of products intended for administration to humans [30,31].

In order to overcome such problems, the use of recombinant aprotinin (rA) for labeling with 99mTc has been proposed [32]. Apronexin™ (produced by Large Scale Biology Corp. Vacaville, CA, USA for Sigma Aldrich (A6103 Sigma Aprotinin-BioUltra)), is manufactured by the transient expression of the aprotinin message in RNA (+)-strand tobacco mosaic virus vectors propagated in non-transgenic Nicotiana plants.

Attempts to label this rA with ^99m^Tc gave satisfactory results with an RCP >95% and stability for up to 24 h after labeling.

To test affinity for synthetic h-insulin fibrils, a saturation binding curve showed a Bmax of 1600 nmol/mg fibrils and a Scatchard plot indicated a Kd of 9.9 × 10^−8^ mol. Competitive binding with cold rA showed a 50% displacement at 102 nmol.

These results, even if not yet verified in vivo, confirm affinity for amyloid fibrils equal to or slightly better than those obtained with bovine A (Table 4) and point towards the obviation of the limited availability of and safety concerns over the vector protein.

### 8.2. Cyclic Peptide 30–51 SS

In order to overcome not only the potential risk of virus and prion contamination of extractive aprotinin, but also the potential Adverse Drug Reactions of r-A and to select a vector compound with faster blood and normal tissue clearance suitable for labeling with a shorter life PET nuclide, the synthesis of an aprotinin analogue was proposed.

Within a MURST framework (Ministry of University and Scientific and Technological Research, Tema 6 PRN Oncologia “Miglioramento dei supporti alla diagnostica per immagini” Project Code 190H67) and in collaboration with Technogen (Piana di Monteverna-Caserta-I), a series of five aprotinin-like peptides were synthetized: 14–38 linear, 30–38 linear and cyclic, and 30–51 linear and cyclic (Figure 1). All these synthetic peptides showed an affinity for amyloid fibrils and the analogue 30-51 SS cyclic (m.w. 2481) showed the best characteristics as far as affinity and suitability to direct (Sn Cl_2_) technetium labeling were concerned. In fact, an RCP > 95% was obtained with a stability of about 3 h. Scatchard analysis of the binding affinity between 30–51 SS and synthetic h-insulin fibrils revealed a Kd of 5.2 × 10^−12^ mol, which is very different to that of Tc-rA (Table 4). Fifty per cent displacement was observed in the presence of Log pmol 2.8/mg fibrils of cold peptide and >8 Log pmol/mg of cold native aprotinin [33].

The affinity constant and low m.w. suggest that in vivo use can offer faster blood and non-target tissue clearance via the kidney, lower hepatic activity leading to higher affinity, and better contrast for organ amyloid deposits. More importantly, affinity for amyloid fibrils is increased by a factor of 10^4^ in comparison with the whole protein, rendering this labeled compound a significant potential breakthrough in the molecular diagnosis of cardiac amyloidosis.

In addition, its peptidic nature is suitable for ^68^Ga labeling via a bifunctional chelator.

### 8.3. Ga-NOTA-Aprotinin

Nuclear medicine technology is rapidly moving toward PET replacing single photon examinations because of its better spatial resolution and the possibility of quantifying the true organ/lesion uptake.

With the aim of translating the results obtained with 99mTcA to a PET methodology, recombinant aprotinin (rA) was conjugated with the intermediate chelator p-SCN-Bz-NOTA (N) to obtain a conjugate (AN). The AN was purified by filtration on an ultracel-PL membrane and labeled with ^67^Ga chloride (rANGa), demonstrating excellent stability (up to 83% after 60 min) and a labeling yield ranging from 73 to 88%.

The labeled compound was tested in a binding assay with synthetic r h-Insulin fibrils. Scatchard analysis of the binding affinity revealed a Kd of 8.59 × 10^−8^ mol. (Table 4 and Figure 2). Competition binding tests showed a 50% displacement at 129 nmol [30].

Even if the labeling yield is still suboptimal, rANGa shows a similar affinity to that of the Tc-labeled compound, indicating no loss of binding properties after conjugation with p-SCN-Bz-NOTA, and may be considered a new potential PET radiopharmaceutical with high affinity for amyloid deposits.

## 9. Labeled Aprotinin in the Present Amyloid Cardiac Imaging Landscape

There is a need for a diagnostic radiopharmaceutical for CA which can fulfill the following requirements: (a) readily available, non-invasive, able to quantify the amyloid burden, and easily reproducible; (b) sufficiently sensitive to reveal subtle amounts of fibrils providing an early diagnosis, and (c) adequate for assessing progression or responses to therapy. This last characteristic aroused great interest following the introduction of new drugs able to modify cardiac involvement [35,36]. More than 30 proteins are present in the fibril-associated amyloid, with AL and ATTR being the most common. Therefore, another question arises—namely, whether a radiopharmaceutical with approximately the same sensitivity for all types of amyloid or one which is more selective for a specific type is preferable for assisting in the diagnostic workup.

In the present cardiac amyloid imaging landscape, technetium-labeled phosphonates, especially -DPD, and -PPi in US, are the most commonly employed tracers. They preferentially detect ATTR-associated cardiac amyloid rather than AL, but lack the ability to quantify disease burden over time and to correlate with serum biomarkers [37]. Despite the fact that the vast majority of patients studied with TcA were affected by AL amyloidosis, positive cardiac scans were obtained also in ATTR and AA—thus confirming the in vitro experience with different amyloid fibrils [9].

However, the binding mechanism of labeled aprotinin is highly specific for fibrils and very different from that observed with ^99m^Tc-DPD and –PPi, which bind amorphous precipitates with the same kinetics of fibrils [38]. More recently, Thelander et al. demonstrated that the binding of bone tracers to amyloid-containing hearts depends on an irregular presence of clouds of very tiny calcifications, which seem not to be directly associated with amyloid fibrils [39].

The introduction of PET radiopharmaceuticals such as ^11^C-PIB, ^18^F-florbetapir and ^18^F-florbetaben has changed the molecular imaging approach. They are potentially able to detect early amyloid deposition and to correlate with disease progression. They preferentially bind to AL fibrils in comparison to ATTR [36]. These tracers demonstrate a preferential affinity for AL rather than ATTR, but the binding site is different from that of aprotinin—which is not displaced by thioflavin in vitro and is stably retained in vivo in amyloid deposits, while PET tracers undergo a washout process [9,36]. This may be a further advantage of aprotinin being either Tc or Ga-labeled, allowing a wider imaging-window time with a unique limit of radioactive decay.

There is only one tandem study by the Japanese group comparing the performance of ^11^C-PIB PET and TcA SPECT/CT in nine AL patients [40]. Disagreement was reported in five of them, with four apparently false positive TcA scans. From a so-limited number of cases, and for the reasons mentioned above relating to possible technical artifacts with TcA SPECT, it is difficult to draw reliable conclusions.

^124^I-peptide p5 + 14 is a new emerging PET pan-amyloid radiopharmaceutical, binding via electrostatic interactions to electronegative glycosaminoglycans (GAGs) and protein fibrils—two ubiquitous components of amyloid deposits. Pathological heart activity, either AL or ATTR, increases up to 6 h and remains almost stable for up to 24 h. Uptake correlates with serum NT-proBNP in AL but not in ATTR patients [41,42]. Potential limits relate to the 124 I label (half-life 4.2 days)—with a restricted availability—in vivo dehalogenation with accumulation of free iodine in the stomach, thyroid, and salivary glands, and the elapsed time after injection required for imaging (6 h).

The peptidic approach suggests once again the use of a 30-51SS cyclic peptide. Due to its peptidic structure, it is suitable for Ga labeling via a bifunctional chelator with or without a spacer and, theoretically, could be a PET tracer with a higher affinity and a faster clearance than the parent molecule.

## 10. Concluding Remarks

TcA was a promising radiopharmaceutical for the detection of cardiac amyloidosis with high sensitivity and specificity supported by a robust biochemical basis. Unfortunately, the temporary suspension of Trasylol and concerns over the risk of viruses and prions in the bovine protein halted further clinical studies.

As recently pointed out by Slart et al. in an editorial published in JNC, attempts to label Aprotinin with positron emitters may be desirable in order to overcome the spatial limitations of planar/SPECT detection and to improve its imaging accuracy [43].

Even if not yet tested in humans, it has been demonstrated that the use of recombinant aprotinin instead of the native protein of bovine origin and labeling with Ga through a bifunctional chelator do not affect binding affinity. In addition, the use of a peptide fragment increases the affinity for amyloid fibrils in a very impressive manner. Therefore, further studies of this molecule and its synthetic peptide appear worthwhile in animal and human models.

## Figures and Tables

**Figure 1 biomedicines-10-01377-f001:**
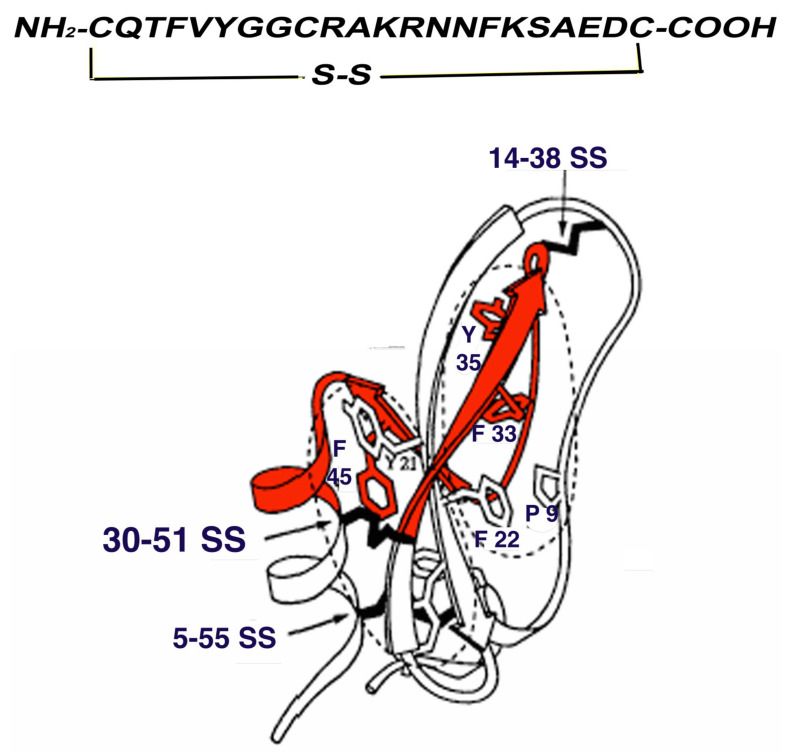
Peptide 30–51 SS (in red) comprising a large part of the second and third β-sheets and part of the α-helix of Aprotinin. Aminoacidic sequence at the top of the figure.

**Figure 2 biomedicines-10-01377-f002:**
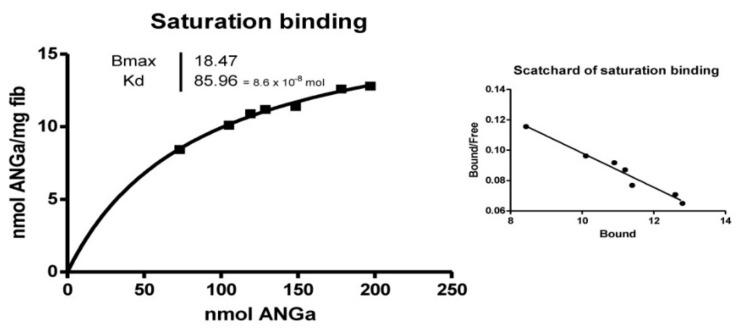
Saturation binding and Scatchard plot of Ga-labelled NOTA-rAprotinin (ANGa) with h-r Insulin fibrils. The labeling procedure is rapid, starting from a cold kit, and is convenient for the positron emitter ^68^Ga (half-life 68 min) obtained from a Ge/Ga generator.

**Table 1 biomedicines-10-01377-t001:** Comparison between Tc-A scans and endomyocardial biopsy results in 28 subjects. Data recorded from refs. [3,17,18,19,20].

		Endomyocardial Biopsy	
		+	−
Tc-A scan	+	16	0
	−	1	11

Table also includes necroscopy results. Note that the Awaya et al. results [20] refer only to planar scans.

**Table 2 biomedicines-10-01377-t002:** Long term follow-up of 72 patients with AL amyloidosis in relation to the pre-test suspicion of myocardial involvement and Tc-Aprotinin scan results. Data from [19].

Pre-Test Score (n)	Evaluable	Scan	Alive	Non Cardiac Death	Cardiac Death	CF Onset or Prog.	Lost at f.u.
A (34)	32	−	26	3			2
		+	1		1	1	
B (15)	13	−	2	2	2		1
		+		1	3	3	1
C (12)	12	−	1	1			
		+		5	1	4	
D (17)	15	−					
		+		4	8	3	2

Pre-test score: A—no suspicion, B—suspected, C—high probability or clear evidence without signs of congestive heart failure (CHF), D—as in C but accompanied by CHF. −/+: cardiac scan negative/positive. CF: Cardiac Failure.

**Table 3 biomedicines-10-01377-t003:** Figures of merit of cardiac amyloid detection with 99mTc-Aprotinin scans.

Scan Results				Subjects under Study	[Ref]
TP	FP	TN	FN		
36	1	35	2	n.. 74 AL biopsy-proven and long-term f.u. (3 localized, 2 MGUS)	[19]
10	-	12	-	n. 22 with known or suspected amyloidosis (14 AL, 2 hereditary form, 3 localized, 3 MM)	[17]
18	-	17	-	n. 35: n.18 biopsy-proven (14 AL, 3 AA, 1 ATTR), n 17 control.	[18]
3	-	6	1	n. 10 patients with biopsy-proven primary amyloidosis	[21]
-	-	12	-	n. 12 control subjects with cardiac non-amyloidosis pathology	[3] ^
4	-	5	1	n. 10: 5 AL type, 5 non-amyloidosis patients, planar scans	[20] *
5	3	2	-	Results of the same study with SPECT/CT	
6	-	4	-	n. 10 subjects with familial ATTR	[22] ¶

^ Data extrapolated from Ref. [3]; * Data from Awaya [20] are reported as obtained with planar scans (upper row) or with SPECT/CT (lower row); ¶ data extrapolated from Ref. [22].

**Table 4 biomedicines-10-01377-t004:** Binding assay with synthetic r h-Insulin fibrils (Kd) of the compounds cited in this paper.

Label	Compound	Kd	Fibrils	[Ref]
^125^I	Aprotinin native	2.9 × 10^−6^ mol	h r-insulin	[9]
^99m^Tc	r-Aprotinin	9.9 × 10^−8^ mol	h r-insulin	[32]
^99m^Tc	30-51SS	5.2 × 10^−12^ mol	h r-insulin	[33]
^67^Ga	r-ANGa	8.59 × 10^−8^ mol	h r-insulin	[34]

r: recombinant; 30-51 SS: cyclic synthetic peptide 30-51 SS; r-ANGa: ^67^Ga labelled NOTA r-Aprotinin.

## Data Availability

Not applicable.

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
