# Peer review of "A Narrative Review of ^99m^Tc-Aprotinin in the Diagnosis of Cardiac Amyloidosis and a New Life for an Unfairly Abandoned Drug"

_biomedicines, 2022, doi:10.3390/biomedicines10061377_

Round 1

Reviewer 1 Report

In this manuscript, Carlo Aprile and Lorenzo Lodola. have reviewed  99mTc-Aprotinin in the diagnosis of cardiac amyloidosis. This is a very well organized and comprehensively covered all aspects of the topic. Manuscript is very well written and clearly presented. Most importantly, the author have put together all the unanswered questions and presented all the limitations. Section 8. Future direction is particularly very well written. This manuscript deserve publication with some minor language corrections. 

Author Response

We thank for your comment. The paper has been submitted to English revision.

Reviewer 2 Report

Aprile et al. presented this manuscript about the detection of cardiac amyloidosis using 99mTechnetium labeled Aprotinin. They highlighted the obstacles to its clinical use and made the case for using its synthetic peptidic fragment as a potential candidate for PET detection of cardiac amyloidosis.

I had to read the manuscript multiple times to understand the structure of the titles and subtitles. The transition from one subtitle to the next seemed unnatural. Otherwise, this document serves as a great summary of current evidence-based knowledge. 

Author Response

We added some sentences in section 8 to explain the passage to subtitles, in this way we hope that the transition from an item to the following can be more progressive.

Reviewer 3 Report

The review concerns the use of TcA in diagnostics of cardiac amyloidosis. The authors discuss in detail its properties and the data on potential application from the limited research conducted so far. This information is presented in a clear and structured way.

Currently available methods - CMR as well as PYP and DPD-scintigraphy - enable diagnosis in most cases. Therefore, I believe that the work requires highlighting why, according to the authors, there is a need to search for new tracers. This is discussed to some extent in point 9. However, I think that such a justification should already be found in the introduction.

Most reviews focus on the advantages of current imaging modalities in cardiac amyloidosis. Listing their limitations and describing the potential possibilities of TcA in such context would make the manuscript more valuable

Author Response

  1. We added in the introduction some sentences focusing the unmet needs of a non-invasive imaging test.
  2. Although the aim of this paper was not to make a comparison of the other nuclear imaging methods in comparison to labelled Aprotinin, we tried to underline the differences with the other radiopharmaceuticals not so much for the clinical results but for the different mechanism of binding and kinetics.